# Intersectional Two-sided Fairness in Recommendation

## ABSTRACT

Fairness of recommender systems (RS) has attracted increasing attention recently. Based on the involved stakeholders, the fairness of RS can be divided into user fairness, item fairness, and two-sided fairness which considers both user and item fairness simultaneously. However, we argue that the intersectional two-sided unfairness may still exist even if the RS is two-sided fair, which is observed and shown by empirical studies on real-world data in this paper, and has not been well-studied previously. To mitigate this problem, we propose a novel approach called *Intersectional Two-sided Fairness Recommendation* (ITFR). Our method utilizes a sharpness-aware loss to perceive disadvantaged groups, and then uses collaborative loss balance to develop consistent distinguishing ability for different intersectional groups. Additionally, predicted score normalization is leveraged to align positive predicted scores to fairly treat positives in different intersectional groups. Extensive experiments and analyses on three public datasets show that our proposed approach effectively alleviates the intersectional two-sided unfairness and consistently outperforms previous state-of-the-art methods.[1]

## 1 INTRODUCTION

As recommender systems (RS) involve the allocation of social resources, the fairness of RS has attracted increasing attention [6, 37, 57]. Fairness in RS can be divided into three types based on the involved stakeholders: user fairness, item fairness, and two-sided fairness which aims to ensure both user and item fairness concurrently. Presently, user fairness mainly entails consistent recommendation performance across different user groups [10, 45], while item fairness primarily focuses on fair exposure [13, 37, 43] or consistent recommendation performance for different item groups [4, 57]. Two-sided fairness is achieved when both user fairness and item fairness criteria are met simultaneously [5, 46].

However, we argue that a form of intersectional two-sided unfairness may still exist even when two-sided fairness is achieved. As illustrated in Fig.1, we present a toy example. Consider a movie recommendation scenario with 200 users, comprising 100 male and 100 female users, and a movie collection consisting solely of horror and romance genres. Suppose the RS recommends only one movie for each user. Among the male users, 90 prefer horror movies, while the remaining 10 favor romance. Conversely, among the female users, 90 prefer romance movies, and the remaining 10 prefer horror. Now, let us examine a straightforward RS strategy that exclusively recommends horror movies to men and romance movies to women. This recommendation strategy adheres to current two-sided fairness criteria, but the intersectional two-sided groups (Female like Horror movies and Male prefer Romance movies) experience discrimination. While this example is simplified, it can be readily extended to accommodate varying user counts, recommendation lengths, and fair distribution. Such a phenomenon has been observed in the real scenario, which is shown and discussed in Section 3.

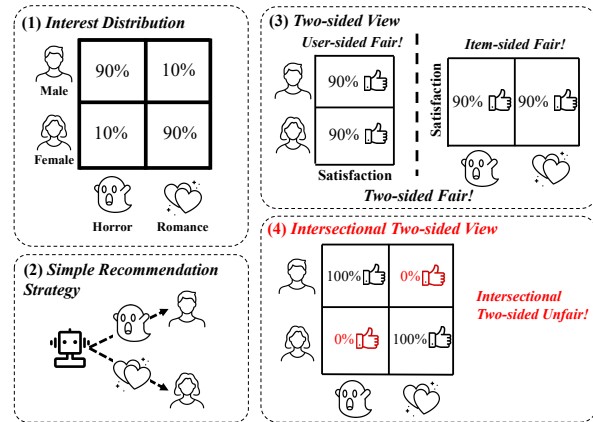

**Figure 1: Illustration of intersectional two-sided unfairness. In this toy example, the RS strategy meets two-sided fairness but shows unfair in an intersectional two-sided view. The thumb-up means the recommendation fits the user interest.**

Such intersectional two-sided unfairness has manifold harm. From the user perspective, some of the user's interests are systematically discriminated against, which may harm recommendation diversity and lower user satisfaction. From the item perspective, the RS fails to explore the potential diverse users for items, potentially burying valuable items. From the platform perspective, this constrains the development of a diverse user-item ecosystem, impeding the platform's progress. Moreover, from a social perspective, such unfairness may reinforce the social polarization issue [32]. Hence, addressing intersectional two-sided unfairness is crucial for RS.

To verify the existence of such unfairness, we conduct empirical experiments in real-world data. We find that intersectional unfairness indeed exists and cannot be ignored. Unfortunately, we further observe that current fairness methods cannot effectively mitigate such unfairness, highlighting the importance of designing approaches to address this problem.

To fill this gap, we design a novel method *Intersectional Two-sided Fairness Recommendation* (ITFR) to mitigate such unfairness. Considering that the positives of an intersectional group compete not only with all the negatives but also with positives from other groups in the same recommendation list, we divide the intersectional two-sided fairness into two goals: (i) consistently distinguishing between positives and negatives for different intersectional groups; (ii) fairly treating positives in different intersectional groups. To achieve the first goal, ITFR employs loss balance, as training losses may be a proxy of the distinguishing ability. However, low training losses do not necessarily indicate poor distinguishing ability, and the training losses of various groups can influence each other. Direct reweighting losses based on their size may not be an effective solution. To tackle the first challenge, ITFR incorporates a sharpness-aware loss to improve the alignment between training losses and test performance, thereby enhancing the identification of disadvantaged

---

[1]Our work is related to the "User Modeling and Recommendation" track as it can improve fairness for recommendation. We will release the codes upon acceptance.

groups. To address the second challenge, ITFR leverages group collaboration information to learn fair weights for intersectional groups, thereby balancing their sharpness-aware losses. Additionally, to achieve the second goal, predicted score normalization is leveraged to align predicted scores for positives. To demonstrate the effectiveness, we conduct extensive experiments on three public datasets. Experimental results show that our method can effectively alleviate the intersectional two-sided unfairness. Our main contributions can be summarized as follows.

- To the best of our knowledge, it is the first work to study the intersectional two-sided fairness in Top-N recommendation. We conduct empirical experiments to show the existence of such unfairness and inadequacy of current fairness methods.
- We propose a novel method ITFR to mitigate the intersectional two-sided unfairness, which consists of *sharpness-aware disadvantage group discovery*, *collaborative loss balance*, and *predicted score normalization*.
- Extensive experimental results on three public datasets demonstrate that our method can effectively mitigate intersectional two-sided unfairness with similar or even better accuracy.

## 2 RELATED WORK

### 2.1 Single-sided Fairness in Recommendation

Below we introduce the two single-sided fairness in recommendation: user fairness and item fairness.

Current research on user fairness can be roughly divided into two groups: learning fair user representations [23, 48] and producing fair recommendation outcomes [10, 14, 21, 22, 45, 54]. The former is related to process fairness, while the latter focuses on the fairness of recommendation performance received by different users, which has attracted more attention as it is more related to user satisfaction. These fairness methods on outcome fairness can be roughly grouped into three categories: (i) fairness regularization [14, 21, 54]. (ii) distributionally robust optimization [45, 52]. (iii) re-ranking [10, 22].

Most existing item fairness studies can be divided into exposure-based fairness (or treatment-based fairness) and performance-based fairness (or impact-based fairness). Exposure-based item fairness focuses on allocating fair exposure to each item [11, 12, 20, 50]. Most of them propose integer programming-based re-ranking methods [2, 13, 24–26, 28, 37–39, 53, 56]. Unlike exposure-based item fairness, performance-based item fairness focuses on whether different item groups have consistent recommendation performance (e.g., recall of the positives), which is related to users' true preferences. The fairness methods on performance-based fairness can be coarsely divided into three categories: (i) fairness regularization [1, 15]. (ii) adversarial learning [57]. (iii) fairly negative sampling [4].

The above methods only enhance single-sided fairness. However, since the RS is a typical two-sided platform, it is important to ensure both user and item fairness.

### 2.2 Two-sided Fairness in Recommendation

Current work [5, 29, 30, 46, 47, 49] on two-sided fairness in Top-N recommendation is aimed to ensure user and item fairness simultaneously. Specifically, most studies focus on ensuring performance-based user fairness (i.e., different users receive consistent recommendation performance) and exposure-based item fairness, except

for [47] focusing on purely exposure fairness in a stochastic ranking scenario. Most work [5, 29, 30, 49] designs fair re-ranking methods to achieve this goal as the allocation of exposure is more feasible in the re-ranking stage, while [46] propose a multi-objective optimization approach in the ranking stage. Unlike these studies, we focus on performance-based fairness both for users and items, which will be further explained in Section 3.1. A similar study is [40], which focuses on the marketing bias in the rating prediction task and also involves intersectional groups. However, the intersectional unfairness in this paper may not be due to the marketing bias, and their method is not designed for Top-N recommendation.

Unlike current work, we argue that ensuring user and item fairness simultaneously is insufficient. This paper aims to alleviate the intersectional two-sided unfairness in Top-N recommendation, which current fairness methods may overlook.

### 2.3 Intersectional Fairness in Machine Learning

There have been several studies [8, 9, 16–18, 34, 41, 51] on intersectional fairness in machine learning (ML), which focuses on the intersectional groups of different attributes, such as race & gender (e.g., black females). These studies argue that when multiple fairness-aware attributes exist, each intersectional group should be treated fairly. Multiple attribute divisions may lead to more sparse and unbalanced subgroups compared to a single attribute, which is the concern of these studies.

Different from these studies on intersectional fairness in ML, we focus on the intersectional two-sided fairness in the Top-N recommendation, which has a key difference: the two-sided group makes it more challenging than the single-sided group, i.e., the single-sided fairness methods and current two-sided fairness methods ignoring the intersectional groups is not designed to mitigate such intersectional two-sided unfairness effectively. However, the intersectional single-sided fairness (e.g., the unfair recommendation performance of black females) might be effectively alleviated by current adequate single-sided fairness methods, as we can just treat the intersectional single-sided groups as a new group division.

## 3 PROBLEM DEFINITION AND EMPIRICAL STUDY

### 3.1 Problem Definition

Suppose there are $n$ users $\mathcal{U} = \{u_1, ..., u_n\}$ and $m$ items $\mathcal{V} = \{v_1, ..., v_m\}$. The collected user feedback can be represented by $\mathcal{Y} \in \{0, 1\}^{n \times m}$, where $y_{ui}$ denotes whether the user $u$ has interacted with the item $i$. The whole positives are $\mathcal{D} = \{(u, i)|y_{u,i} = 1\}$. Ideally, there exists an unobserved matrix $\mathcal{R} \in \{0, 1\}^{n \times m}$, where $r_{ui}$ represents whether a user $u$ will interact with an item $i$. The top-N recommendation task is to recommend a list of $N$ uninteracted items to each user $u$.

In this paper, we focus on group-level fairness. Suppose the users and items are divided into $P$ and $Q$ disjoint groups by some predefined attributes, respectively. As each interaction belongs to a user group and an item group simultaneously, the whole data $\mathcal{D}$ consist of $P \times Q$ disjointed intersectional two-sided groups.

To study the fairness of these intersectional two-sided groups, we further define the utility of these groups. Specifically, the utility

of an intersectional two-sided group should reflect the received recommendation performance of potential interactions in this group. Formally, let $\mathcal{U}(i, j)$ denote all users in the $i$-th user group who are interested in at least one uninteracted item in the $j$-th item group. The utility of the intersectional two-sided group $(i, j)$ is defined as the average utility for these potential interests:

$$\text{ITG\_Utility}(i, j)@K = \frac{1}{|\mathcal{U}(i, j)|} \sum_{u \in \mathcal{U}(i, j)} \text{utility}(u, j)@K \quad (1)$$

where utility$(u, j)@K$ can be some metrics measuring recommendation performance. Without loss of generality, we follow previous work [4, 57] and use a recall-based metric, i.e., utility$(u, j)@K = \frac{|\{i|i \in l_u \& r_{u,i}=1 \& i \in \mathcal{V}_j\}|}{|\{i|y_{u,i}=0 \& r_{u,i}=1 \& i \in \mathcal{V}_j\}|}$, here $l_u$ is the top-K recommendation list for user $u$. Note that we ignore users not interested in the $j$-th item group, as the utility for these users is always zero and meaningless.

Based on the utility definition, intersectional two-sided fairness aims to provide similar utilities for different groups.

The reason to choose such a performance-based utility definition instead of an exposure-based utility definition (e.g., the received exposure for intersectional groups) is that the latter does not consider user preferences. Specifically, for user fairness, the exposure-based utility is inconsistent with current user fairness that focuses on recommendation performance related to user preferences [22, 45]. For item fairness, it is also crucial to consider user preferences [4]. Only exposure-based item fairness without performance-based fairness might cause some item groups to receive low recommendation quality, i.e., recommended to the users uninterested in them [44].

## 3.2 Existence of Intersectional Unfairness

Below we investigate whether such unfairness exists in real datasets. For brevity, we only use a classic dataset Movielens1M (ML1M) to conduct our empirical experiments. Here we only consider the binary group setting, and in subsequent experimental sections, we show results for more than two groups. We use gender to divide users (Male v.s. Female) and movie genres to divide items (here we take 'Children's v.s. Horror' as an example). The processed dataset ML1M-2 contains 4,403 users, 568 items, and 144,420 interactions. We randomly divide all interactions into training, validation, and test sets in the ratio of 7:1:2. We run the classic BPR [33] algorithm and repeat it five times, and the results are shown in Table 1, where URecall@20 is the average Recall@20 for the user group, and IRecall@20 is the recall at the item group level [57].

**Table 1: Results of BPR (ML1M-2).** *Italic* for the bottom two intersectional groups and the worst single-sided group.

| ITG_Utility@20 | | User | | IRecall@20 |
| | | Female | Male | |
|---|---|---|---|---|
| Item | Children's | 0.5215 | *0.4669* | 0.4440 |
| | Horror | *0.4125* | 0.4814 | *0.4101* |
| URecall@20 | | 0.5070 | *0.5018* | - |

From the single-sided fairness perspective, we can find that 'Children's' gets a significantly better performance in terms of item fairness, while male and female users get very similar performance

without significant differences. Existing two-sided fairness only requires single-sided fairness for both users and items. Therefore, the model should improve the performance of 'Horror' in terms of item fairness while keeping the current fair status on the user side.

However, the story is different from the intersectional two-sided perspective. As shown in Table 1, these four intersectional two-sided groups receive inconsistent recommendation quality, with (Male & Children's) and (Female & Horror) receiving worse performance. The best group (Female & Children's) has an about 26% performance gap compared to the worst group (Female & Horror), which indicates that intersectional two-sided unfairness indeed exists. Note that the two best intersectional groups are on the diagonal, which is consistent with Fig.1.

## 3.3 Do Current Methods Help?

Next, we investigate the effectiveness of current fairness methods. We use two advanced single-sided fairness methods: FairNeg [4] for item fairness and StreamDRO [45] for user fairness, which both are focused on performance-based fairness and applied in the ranking stage. Although there is no performance-based two-sided fairness method, for experimental completeness, we use an in-processing two-sided fairness method MultFR [46] which focuses on performance-based user fairness and exposure-based item fairness.

**Table 2: Results of FairNeg (item fairness) on the ML1M-2 dataset.** *Italic* for the bottom two intersectional groups and the worst single-sided group. The (↑/↓) means better or worse recommendations compared with BPR in Table 1.

| ITG_Utility@20 | | User | | IRecall@20 |
| | | Female | Male | |
|---|---|---|---|---|
| Item | Children's | 0.5042(↓) | *0.4490(↓)* | 0.4281(↓) |
| | Horror | *0.4258(↑)* | 0.4956(↑) | *0.4266(↑)* |

**Table 3: Results of StreamDRO (user fairness) on the ML1M-2 dataset. The notations are similar to Table 2.**

| ITG_Utility@20 | | User | |
| | | Female | Male |
|---|---|---|---|
| Item | Children's | 0.5202(↓) | *0.4696(↑)* |
| | Horror | *0.4121(↓)* | 0.4816(↑) |
| URecall@20 | | 0.5066(↓) | *0.5031(↑)* |

As shown in Tables 2 and 3, current single-sided fairness methods indeed improve targeted single-sided fairness. However, they do not improve intersectional two-sided fairness very well. For item fairness, FairNeg indeed narrows the overall gap between item groups, improving item fairness in the single-sided view. However, in the intersectional view, it improves the performance for all the Horror groups, leading to better performance for some advantaged groups (Male & Horror) and worse performance for some disadvantaged groups (Male & Children's). For user fairness, a similar phenomenon can be found in Table 3, where some advantaged groups (Male

Table 4: Results of MultiFR (two-sided fairness) on the ML1M-2 dataset. The notations are similar to Table 2.

| ITG_Utility@20 | | User | | IRecall@20 |
|---|---|---|---|---|
| | | Female | Male | |
| Item | Children's | 0.5151(↓) | *0.4671(↑)* | 0.4394(↓) |
| | Horror | *0.4072(↓)* | 0.4809(↓) | *0.4086(↓)* |
| URecall@20 | | 0.5019(↓) | *0.5016(↓)* | - |

& Horror) receive better recommendations and some disadvantaged groups (Female & Horror) receive worse recommendations.

The results for two-sided fairness methods are shown in Table 4. We can find that it indeed narrows the gap between different user groups, but does not effectively improve performance-based item fairness as it considers exposure-based fairness. It can also be found that the worst intersectional group (Female & Horror) in Table 1 receives worse recommendations.

As current methods cannot effectively mitigate such unfairness, it is important to design an effective fairness method for improving intersectional two-sided fairness.

## 4 INTERSECTIONAL TWO-SIDED FAIRNESS RECOMMENDATION

### 4.1 Overview

The utility of an intersectional group is determined by the rank of the positives in this group in the recommendation lists. These positives compete with two kinds of samples: all the negatives and other positives from distinct groups in the recommendation list. Thus, we divide the intersectional two-sided fairness into two goals to balance these competitions separately. As shown in Fig.2, (i) the RS should consistently distinguish between positives and negatives for different intersectional groups; (ii) the RS should treat positives in different intersectional groups fairly to ensure that no positives in a certain group have systematically low predicted scores.

To achieve the above two goals, we propose a method *Intersectional Two-sided Fairness Recommendation* (ITFR), which consists of three components: *sharpness-aware disadvantage group discovery*, *collaborative loss balance*, and *predicted score normalization*. The purpose of the first two components is to balance the training losses between different intersectional groups, which reflects the ability to distinguish between positives and negatives, corresponding to the first goal. Nevertheless, low training losses do not necessarily indicate poor test performance, and different intersectional groups are related to each other. Direct reweighting losses based on their size may not be an effective solution. To tackle the first challenge, we introduce *sharpness-aware disadvantage group discovery* to enhance the consistency of training losses and test performance. To address the second challenge, we leverage the group collaboration information to learn fair weights for these intersectional groups, i.e., *collaborative loss balance*.

However, only controlling the training loss may not meet the second goal. Even if the training loss is similar between different intersectional groups, the predicted score for positive samples in different intersectional groups may be systematically biased as the

commonly used recommendation loss (e.g., BPR) only optimizes the distance between positives and negatives and does not constrain the absolute value of the predicted scores. Therefore, the third component, *predicted score normalization*, is applied to achieve the second goal by aligning positive predictions. Next, we elaborate on our method from the above three components respectively.

### 4.2 Sharpness-aware Disadvantage Group Discovery

Let us first consider the first goal, i.e., to fairly distinguish between positives and negatives for different intersectional groups. First, we need to perceive those intersectional groups with poor distinguishing ability on the test data. Since test data is not available, the intuitive idea is to treat the training loss as a proxy for distinguishing ability on the test data, as they mostly reflect the ability of RS to distinguish positives and negatives. Higher training loss is likely to represent poorer distinguishing ability.

Below we formalize the training loss of the intersectional groups. Take the most commonly used recommendation loss BPR [33] as an example. The BPR loss $\mathcal{L}_{p,q}$ for an intersectional group $g_{p,q}$ is defined as follows:

$$\mathcal{L}_{p,q}(\mathcal{D};\theta) := \frac{1}{|\tilde{\mathcal{D}}_{p,q}|} \sum_{(u,i,j) \in \tilde{\mathcal{D}}_{p,q}} \text{BPR}(u,i,j) \quad (2)$$

Here $\tilde{\mathcal{D}}_{p,q} = \{(u,i,j) | u \in \mathcal{U}_p, i \in \mathcal{V}_q, y_{u,i} = 1, y_{u,j} = 0\}$, $\text{BPR}(\cdot)$ is the BPR loss for a triple pair. $g_{p,q}$ is the intersectional two-sided group corresponding to the $p$-th user group and the $q$-th item group.

Moreover, in addition to the value of training losses, the geometric properties (e.g., sharpness) of the loss around the parameters $\theta$ also impact the test performance [7]. Considering training losses on only a single point $\theta$ is vulnerable to random perturbations if the loss curve is sharp, which may lead to an ineffective detection of discriminated groups. To alleviate this, inspired by related work in machine learning [7], we use the worst training loss within a bounded region of the current parameters $\theta$ as a proxy for the model's distinguishing ability. Formally, the worst loss of the model parameter $\theta$ in the $\rho$-region (i.e., $\{\theta + \epsilon \mid \|\epsilon\| \le \rho\}$) for the group $g_{p,q}$ can be defined as:

$$\hat{\mathcal{L}}_{p,q}(\mathcal{D}, \theta, \rho) := \max_{\|\epsilon\| \le \rho} \mathcal{L}_{p,q}(\mathcal{D};\theta + \epsilon) \quad (3)$$

Compared with the original loss $\mathcal{L}_{p,q}(\mathcal{D};\theta)$, the loss $\hat{\mathcal{L}}_{p,q}$ considers the sharpness of the original loss around the parameters $\theta$ since a larger sharpness will lead to a larger difference between the original loss and the worst loss.

**Practice Detail.** The above definition is not feasible in practice as solving Eq.(3) is time-costing. Following [7], we use single-step gradient ascent to approximate the worst loss. The corresponding solution for Eq.(3) is $\theta_{p,q}^* = \theta + \epsilon_{p,q}^*$, where $\epsilon_{p,q}^* = \rho \frac{\nabla_\theta \mathcal{L}_{p,q}(\mathcal{D};\theta)}{\|\nabla_\theta \mathcal{L}_{p,q}(\mathcal{D};\theta)\|}$.

### 4.3 Collaborative Loss Balance

As the sharpness-aware loss reflects the distinguishing ability, the next question is how to balance the sharpness-aware loss $\hat{\mathcal{L}}_{p,q}(\mathcal{D}, \theta, \rho)$ between different intersectional groups. An intuitive idea is reweighting, i.e., assigning higher training weights to groups with higher losses. The group distributional robustly optimization (GroupDRO)

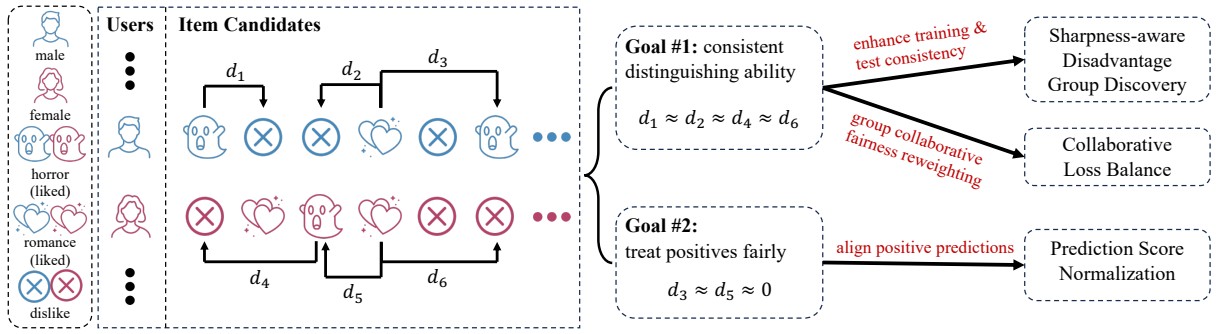

**Figure 2: Illustration for our method.** $d$ denotes the difference in predicted scores between two interactions.

[35] in machine learning can be leveraged to achieve this goal. Specifically, GroupDRO will assign a weight $w_i$ for the $i$-th group, and calculate the total loss as $\mathcal{L}_{GroupDRO} = \sum_i w_i \mathcal{L}_i(\theta)$, where $\mathcal{L}_i(\theta)$ is the average training loss of the $i$-th group, and $w_i$ is updated at each batch by:

$$w_i \leftarrow \frac{w_i \cdot \exp(\eta \cdot \mathcal{L}_i(\theta))}{\sum_j w_j \cdot \exp(\eta \cdot \mathcal{L}_j(\theta))} \qquad (4)$$

where $\eta$ is a hyperparameter. We can adopt the GroupDRO for intersectional two-sided fairness by replacing $w_i$ with $w_{p,q}$ and $\mathcal{L}_i(\theta)$ with $\mathcal{L}_{p,q}(\mathcal{D}; \theta)$. Note that the original GroupDRO does not consider the sharpness of losses. To capture the sharpness, we can directly replace $\mathcal{L}_{p,q}(\mathcal{D}; \theta)$ with $\hat{\mathcal{L}}_{p,q}(\mathcal{D}, \theta, \rho)$.

However, the above method has a drawback as it ignores the collaboration between different intersectional groups, which is important for RS. Typically, there is much collaborative information in users' interactions, and different intersectional groups are related to each other as they may share similar users and items. Therefore, optimizing the loss of one group can also strongly impact the losses of other groups. The current weight $w_{p,q}$ only considers the current group loss $\hat{\mathcal{L}}_{p,q}$ and does not consider the influence between different groups, which may lead to suboptimal performance.

To model the relationship between different intersectional groups, we define the contribution of a group $g_{p,q}$ to $g_{a,b}$ as the change of sharpness-aware loss of $g_{a,b}$ after updating $\theta$ using the sharpness-aware loss $g_{p,q}$, where $\nabla_\theta \hat{\mathcal{L}}_{p,q}$ is the gradients of $\hat{\mathcal{L}}_{p,q}(\mathcal{D}, \theta, \rho)$:

$$\begin{aligned} C(g_{p,q} \to g_{a,b}) &:= \hat{\mathcal{L}}_{a,b}(\mathcal{D}, \theta - \alpha \nabla_\theta \hat{\mathcal{L}}_{p,q}, \rho) - \hat{\mathcal{L}}_{a,b}(\mathcal{D}, \theta, \rho) \\ &\approx \mathcal{L}_{a,b}(\mathcal{D}; \theta^*_{a,b} - \alpha \nabla_\theta \hat{\mathcal{L}}_{p,q}) - \hat{\mathcal{L}}_{a,b}(\mathcal{D}, \theta, \rho) \end{aligned} \qquad (5)$$

Furthermore, the total contribution of an intersectional group is defined as a weighted sum of its contributions to all groups:

$$C(g_{p,q}) = \sum_{a=1}^{P} \sum_{b=1}^{Q} \beta_{a,b} C(g_{p,q} \to g_{a,b}), \text{where } \beta_{a,b} = \frac{(\mathcal{L}_{a,b})^\gamma}{\sum_{p,q} (\mathcal{L}_{p,q})^\gamma} \qquad (6)$$

Here we introduce the group weight $\beta_{a,b}$ because drops in larger losses are more valuable. $\gamma$ is a hyperparameter, and a larger $\gamma$ means we pay more attention to disadvantaged groups.

Given the total contribution $C(g_{p,q})$ of each group $g_{p,q}$, we can calculate the group weight $w_{p,q}$ following Eq.4:

$$w_{p,q} \leftarrow \frac{w_{p,q} \cdot \exp(\eta \cdot C(g_{p,q}))}{\sum_{a,b} w_{a,b} \cdot exp(\eta \cdot C(g_{a,b}))} \qquad (7)$$

The final collaborative balanced loss is $\mathcal{L}_{clb} = \sum_{p,q} w_{p,q} \cdot \hat{\mathcal{L}}_{p,q}$. The total procedure for the first goal can be found in the Appendix.

**Practice Detail.** In practice, Eq.(5) will introduce a high computational cost. Following [31], we use the first-order Taylor approximation and get $C(g_{p,q} \to g_{a,b}) \approx \alpha \nabla_\theta \hat{\mathcal{L}}_{p,q}^T \nabla_\theta \hat{\mathcal{L}}_{a,b}$, the $\nabla_\theta \hat{\mathcal{L}}_{p,q}$ here is further approximated by $\sqrt{\hat{\mathcal{L}}_{p,q}} \frac{\nabla_\theta \hat{\mathcal{L}}_{p,q}}{\|\nabla_\theta \hat{\mathcal{L}}_{p,q}\|}$ to obtain a stable optimization, and the $\nabla_\theta \hat{\mathcal{L}}_{a,b}$ is approximated similarly. Besides, as the parameters are not shared between different users and items in common ID-based RSs, and two groups within a batch may not have overlapped users and items, the $\nabla_\theta \hat{\mathcal{L}}_{p,q}^T \nabla_\theta \hat{\mathcal{L}}_{a,b}$ will be zero as the gradients of two intersectional groups have no overlap in each batch. To alleviate this problem, we use the cumulative gradients of the last epoch as the approximation of $\nabla_\theta \hat{\mathcal{L}}_{a,b}$.

### 4.4 Predicted Score Normalization

Although the proposed loss balance method can improve the first goal, it may not necessarily satisfy the second goal, i.e., to fairly treat positives in different intersectional groups. This is because the commonly used BPR loss only optimizes the distance between positives and negatives. Even if the distances between positives and negatives are the same across intersectional groups, there may still be systematic unfairness in their predicted scores for positives. Note that the competition between positives occurs only between items and that the predicted scores of positive samples are not comparable across users, so this issue may have a more significant impact on item fairness than user fairness.

As directly controlling predicted scores may result in a large accuracy loss [57], we leverage an indirect approach here to alleviate this problem. Note that the proposed loss balance method enhances the similarity of distances between positives and negatives across different intersectional groups. If the range of predicted scores is bounded, then the systematic bias between different intersectional groups may be mitigated, as this bias is restricted to a certain bound rather than over the real number domain. Thus, given user embedding $u$ and item embedding $v$, we bound the commonly used

inner product predicted scores $\hat{y}_{u,v} = u^T v$ to $(-\tau, \tau)$, formally, in an embedding normalization manner:

$$\hat{y}_{u,v} = \tau \cdot \frac{u^T v}{\|u\|\|v\|}. \tag{8}$$

There could be other ways to normalize the predicted scores, e.g., $\hat{y}_{u,v} = \tau \cdot \text{sigmoid}(u^T v)$. However, the embedding normalization manner has its unique advantages: (i) the normalization of user embeddings makes training losses more comparable across users, given that the magnitude of user embeddings influences the training losses but does not affect the recommendation lists. (ii) the normalization of item embeddings may partially alleviate the popularity bias [3], which may be one of the reasons for the predicted score inconsistency between different item groups.

## 5 EXPERIMENTS

### 5.1 Datasets and Settings

*5.1.1 Datasets.* Experiments are conducted on three public datasets: Movielens1M[2], Tenrec-QBA[3] [55] and LFM2B[4] [36].

**Movielens1M.** This dataset contains 1 million movie ratings with user and item profiles. Gender is used to divide user groups, while movie genres are utilized to divide item groups, a commonly used group division in fairness studies [4, 46, 57]. Specifically, we select six genres ('Sci-Fi', 'Adventure', 'Crime', 'Romance', 'Children's', 'Horror') as previously used in [57].

**Tenrec-QBA.** This dataset is collected from a news recommendation platform comprising 348K article clicks. The age is used to divide user groups. Specifically, as the age attribute is grouped in decades with a disrupted order and some decades have little data, we choose the three most popular attribute values ('1', '7', '8'). For item, we use the article channel to divide groups and select the four most popular attribute values ('104', '113', '124', '127').

**LFM2B.** This dataset contains two billion listening events, some of which include genre information. User groups are segmented based on gender. For item, we choose four of the most popular genres with large style differences: ('rock', 'pop', 'jazz', 'ambient').

For all datasets, we remove irrelevant users and items and then randomly divide all interactions into training, validation, and test sets in the ratio of 7:1:2. Statistics of datasets are shown in Table 5.

**Table 5: Statistics of the processed datasets.**

| Dataset | #Users | #Items | #Interactions | Density |
|---|---|---|---|---|
| **Movielens** | 5,977 | 1,200 | 396,207 | 0.0552 |
| **Tenrec** | 11,376 | 1,015 | 132,981 | 0.0115 |
| **LastFM** | 20,847 | 18,625 | 1,785,420 | 0.0046 |

*5.1.2 Metrics.* For accuracy metrics, we adopt the widely used NDCG@K (N@K), Precision@K (P@K), and Recall@K (R@K).

For intersectional two-sided fairness metrics, given the utility definition in Eq.1, let $Util$ denotes the set of all the intersectional group utilities, $Util_{i,:}$ denotes the set of all the intersectional group

[2] https://grouplens.org/datasets/movielens/1m/
[3] https://static.qblv.qq.com/qblv/h5/algo-frontend/tenrec_dataset.html
[4] http://www.cp.jku.at/datasets/LFM-2b/

utilities in $i$-th user group, and similarly, $Util_{:,j}$ denotes the set of all the utilities in $j$-th item group. We use the coefficients of variation [4, 57] to measure unfairness between different groups, i.e., $CV@K = \frac{\text{std}(Util)}{\text{mean}(Util)}$, where $\text{std}(\cdot)$ is the standard deviation and $\text{mean}(\cdot)$ is the average value. We also adopt a metric $MIN@K$ to measure the worst group utility. As the worst utility may be unstable [21], the average utility of the worst 25% groups is measured.

To evaluate single-sided fairness, we also use the coefficients of variation to measure the average unfairness of utilities at the targeted single side: $ICV@K = \frac{1}{P} \sum_{i=1}^{P} \frac{\text{std}(Util_{i,:})}{\text{mean}(Util_{i,:})}$ and $UCV@K = \frac{1}{Q} \sum_{i=1}^{Q} \frac{\text{std}(Util_{:,i})}{\text{mean}(Util_{:,i})}$, measuring item and user fairness, respectively.

*5.1.3 Baselines.* We compare with the following baselines:

- **BPR** [33]: The classic Bayesian personalized ranking method, which does not consider fairness.
- **StreamDRO** [45]: An advanced performance-based user fairness method using a streaming distributionally robust optimization.
- **DPR-REO** [57]: A method for performance-based item fairness using adversarial learning.
- **FairNeg** [4]: An advanced performance-based item fairness method using adaptive fair negative sampling.
- **MultiFR** [46]: An in-processing two-sided fairness method using multi-objective optimization.
- **GroupDRO** [35]: A reweighting method adopted to this problem as Eq.4, which is not originally designed for recommendation. It is a strong baseline as it is aware of intersectional groups.
- **ITFR (ours)**: Our proposed method which uses sharpness-aware collaborative loss balance and predicted score normalization to improve intersectional two-sided fairness.

All the above methods are applied to the ranking phase in RS, and their comparative results are shown in Section 5.2. We also compare with the following two-sided reranking methods:

- **TFROM** [49]: A reranking method to improve exposure fairness for items and balance the performance losses for users.
- **PCT** [42]: An advanced reranking method to improve exposure fairness for items and reduce exposure miscalibration for users.

As these two-sided reranking methods are applied in reranking stage without conflict with our method, we evaluate the compatibility of our method with these methods in Section 5.5. [29] is excluded due to its limited applicability to two-group settings while we are handling multi-group settings.

*5.1.4 Implement Details.* Due to the limited space, more implementation details can be found in the Appendix.

### 5.2 Overall Performance: RQ1

As shown in Table 6, our proposed method ITFR significantly enhances the intersectional two-sided fairness compared to all the baselines, while also maintaining a comparable or even better accuracy. There are some further observations: (i) Firstly, current single-sided fairness methods indeed improve their respective targeted single-sided fairness, even when using more fine-grained fairness metrics, which enables them to alleviate intersectional two-sided unfairness partially. However, it is worth noting that they occasionally exhibit fairness compromises on the other side (e.g.,

Table 6: Performance comparisons. Bold for the best and underline for the second best. */** indicate $p \leq 0.05/0.01$ for the t-test of ITFR vs. the best baseline. ↑/↓ means the higher/lower the better. The improvements are calculated based on the best baseline.

| Dataset | Method | P@20↑ | R@20↑ | N@20↑ | MIN@20↑ | CV@20↓ | UCV@20↓ | ICV@20↓ |
|---------|--------|-------|-------|-------|---------|--------|---------|---------|
| Movielens | BPR | 0.2044 | 0.3793 | **0.3611** | 0.2398 | 0.2026 | 0.0892 | 0.1925 |
| | DPR-REO | 0.2013 | 0.3673 | 0.3502 | 0.2593 | 0.1567 | 0.0906 | 0.1480 |
| | FairNeg | 0.2034 | 0.3761 | 0.359 | 0.2626 | 0.1302 | 0.0953 | 0.1214 |
| | StreamDRO | 0.2043 | **0.3794** | 0.3607 | 0.2389 | 0.204 | 0.0859 | 0.1923 |
| | MultiFR | 0.2033 | 0.3768 | 0.3594 | 0.2403 | 0.1982 | 0.0887 | 0.1876 |
| | GroupDRO | 0.2016 | 0.3757 | 0.3558 | 0.2737 | 0.1145 | 0.0604 | 0.1096 |
| | ITFR(ours) | **0.2074***(+1.4%) | 0.3790(-0.1%) | 0.3605 (-0.1%) | **0.3023**\*\*(+10.4%) | **0.0646**\*\*(-43.5%) | **0.0433**\*\*(-28.3%) | **0.0578**\*\*(-47.2%) |
| Tenrec | BPR | 0.0381 | 0.3493 | 0.175 | 0.2703 | 0.1269 | 0.0453 | 0.1194 |
| | DPR-REO | 0.0378 | 0.3488 | 0.1744 | 0.2764 | 0.1126 | 0.0439 | 0.1048 |
| | FairNeg | 0.0385 | 0.3517 | 0.1756 | 0.2789 | 0.1138 | 0.0445 | 0.1039 |
| | StreamDRO | 0.0382 | 0.3501 | 0.1759 | 0.2721 | 0.1223 | 0.0372 | 0.1154 |
| | MultiFR | 0.0376 | 0.3471 | 0.1735 | 0.2733 | 0.1180 | 0.0429 | 0.1120 |
| | GroupDRO | 0.0378 | 0.3471 | 0.1738 | 0.2711 | 0.1342 | 0.0460 | 0.1248 |
| | ITFR(ours) | **0.0401**\*\*(+4.1%) | **0.3647**\*\*(+3.6%) | **0.1818**\*\*(+3.3%) | **0.3075**\*\*(+10.2%) | **0.0793**\*\*(-29.5%) | **0.0319**\*\*(-14.2%) | **0.0713**\*\*(-31.3%) |
| LastFM | BPR | 0.1090 | 0.1475 | 0.1655 | 0.0538 | 0.3398 | 0.0762 | 0.3369 |
| | DPR-REO | 0.1078 | 0.1426 | 0.1617 | 0.0615 | 0.2989 | 0.0783 | 0.2960 |
| | FairNeg | 0.1095 | 0.1485 | 0.1662 | 0.0694 | 0.2896 | 0.0794 | 0.2866 |
| | StreamDRO | 0.1092 | 0.1478 | 0.1655 | 0.0537 | 0.339 | 0.0704 | 0.3362 |
| | MultiFR | 0.1079 | 0.1462 | 0.1634 | 0.0473 | 0.3687 | 0.0720 | 0.3658 |
| | GroupDRO | 0.1082 | 0.1466 | 0.1642 | 0.0896 | 0.1923 | 0.0650 | 0.1851 |
| | ITFR(ours) | **0.1114**\*\*(+1.7%) | **0.1577**\*\*(+6.1%) | **0.1704**\*\*(+2.5%) | **0.0956**\*\*(+6.6%) | **0.1772**\*\*(-7.8%) | **0.0648**(-0.3%) | **0.1680**\*\*(-9.8%) |

Movielens), a phenomenon discussed in previous work [49]. (ii) Secondly, the two-sided fairness method MultiFR also partially mitigates intersectional two-sided unfairness. Nevertheless, it exhibits instability and does not consistently improve fairness across all datasets, primarily due to its lack of consideration for intersectional groups and its primary focus on optimizing item exposure fairness rather than performance-based fairness. (iii) Thirdly, GroupDRO achieves notably higher fairness compared to the aforementioned fairness methods on the Movielens and LastFM datasets. However, its performance is less satisfactory on the Tenrec dataset. The former can be attributed to its explicit consideration of intersectional groups. The latter observation suggests that focusing solely on loss balance does not necessarily guarantee fairness, as discussed in the method section. (iv) Fourthly, our proposed method consistently outperforms all others in terms of fairness across all datasets, which demonstrates its effectiveness. This superior performance is attributed to its consideration of intersectional groups and its simultaneous pursuit of both two fairness goals. Furthermore, our method incurs only a negligible loss of accuracy on the Movielens dataset and even attains the best accuracy on the Tenrec and LastFM datasets. To conclude, these results validate that our method ITFR can effectively mitigate intersectional two-sided unfairness while maintaining similar or even better accuracy.

## 5.3 Ablation Study: RQ2

### 5.3.1 Ablation for three components.
We conduct ablation studies to assess the effectiveness of the three components within our method. Specifically, three variants are examined: ITFR w/o sharpness-aware loss (SA), ITFR w/o collaborative loss balance (CB) and ITFR w/o predicted score normalization (PN).

As shown in Fig.3, each module demonstrates efficacy in enhancing fairness. The effectiveness of these modules exhibits some

variation contingent upon dataset characteristics. For instance, SA noticeably enhances fairness in the Tenrec dataset, CB is particularly effective in the LastFM dataset, and PN exhibits substantial effectiveness in both the Movielens and Tenrec datasets. As for accuracy, we observe a consistent enhancement across all three datasets with the inclusion of PN, which may be owed to its ability to alleviate popularity bias [3].

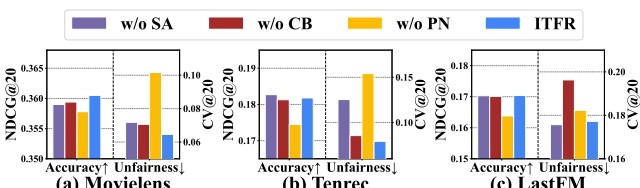

Figure 3: Ablation for three components in our method. "SA": sharpness-aware disadvantage group discovery. "CB": collaborative loss balance. "PN": predicted score normalization.

### 5.3.2 Ablation for two goals.
We further conduct ablation studies to assess the effectiveness of two fairness goals: consistent distinguishing ability (Goal 1) and treating positives fairly (Goal 2). In addition to BPR (None) and ITFR (Goal 1 & Goal 2), we explore two variants: ITFR w/o PN (Goal 1) and BPR w. PN (Goal 2).

The results are depicted in Fig.4. We can find that both goals are important for intersectional two-sided fairness, particularly on the Tenrec dataset, where neither goal in isolation can improve fairness. Besides, Goal 2 alone yields the best accuracy, but it is not effective in improving fairness. Moreover, it enhances overall fairness less effectively than Goal 1, possibly due to the indirect way to improving Goal 2. We leave the exploration of more effective methods to achieve Goal 2 for future work.

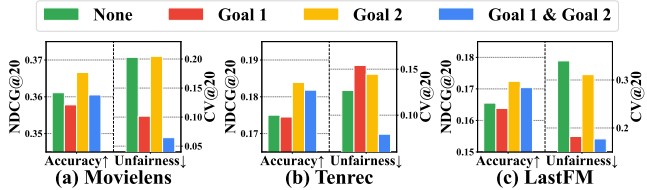

Figure 4: Ablation for two goals in our method.

## 5.4 Hyperparameter Analysis: RQ3

We further analyze the influence of the hyperparameters involved in our method, specifically $\rho$ in SA, $\eta$ and $\gamma$ in CB, and $\tau$ in PN. The results are presented in Fig.5. For all parameters, fairness tends to initially improve and subsequently decline as the parameter value increases. A similar pattern is observed in accuracy, though with some variations in magnitude. For $\rho, \eta, \gamma$, increasing the value within a specific range amplifies the impact of the corresponding component, resulting in further improvements in fairness. However, excessively large values can compromise optimization stability and result in a rapid decline in performance. Regarding $\tau$, it influences the gradient magnitude at each update to some extent; thus, too large or too small values are not suitable.

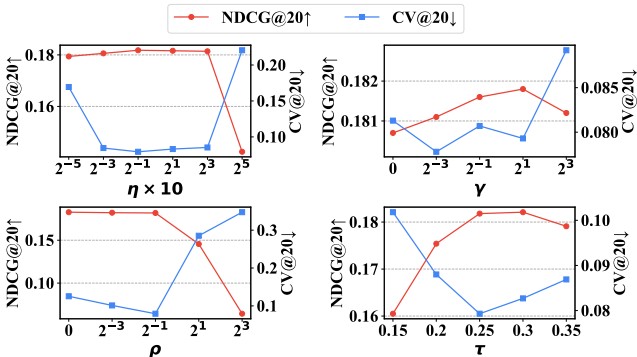

Figure 5: Hyperparameter analysis on the Tenrec dataset. Results on other datasets are similar and omitted.

## 5.5 Compatibility with Reranking Alg.: RQ4

Most current two-sided fairness methods [29, 42, 49] are reranking-based and focus on exposure fairness for items. Since our method is applied to the ranking phase without conflicting with these reranking methods, we next verify its compatibility with these methods. We consider two advanced two-sided fairness-aware reranking methods: TFROM [49] and PCT [42]. Exposure fairness requires a definition of a fair exposure distribution. We follow the previous work [42] that each item group should have equal exposure. For metrics, we utilize the KL-divergence [13, 40] between the fair distribution and the system distribution to measure item exposure unfairness, denoted as $SystemKL$. In addition, following [42], miscalibration ($UserKL$) is used to measure whether users receive recommendations that fairly reflect their historical interests, which can be regarded as exposure fairness at the user level.

As depicted in Fig.6, the results indicate that our method is compatible with these reranking methods. For exposure fairness, utilizing ITFR as inputs can achieve similar or even better fairness, which shows that our method does not disrupt the efficacy of these reranking methods. In addition, for intersectional two-sided fairness, it can be found that reranking has a notable detrimental effect on fairness, but ITFR still exhibits an improvement compared to BPR, underscoring its utility even in the presence of disturbances during the reranking phase.

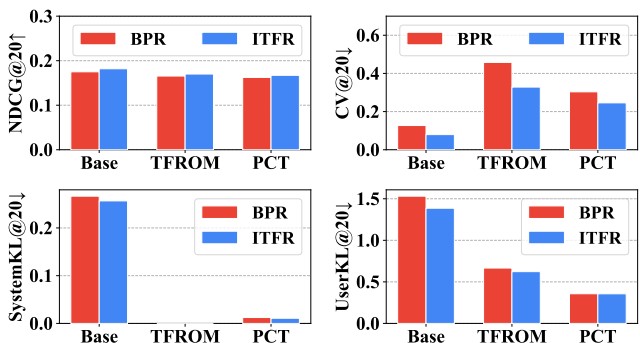

Figure 6: Reranking results on the Tenrec dataset. Results on other datasets are similar and omitted. All metrics are the lower the better except for NDCG@20.

## 6 CONCLUSIONS AND FUTURE WORK

This paper aims to mitigate the intersectional two-sided unfairness in Top-N recommendation, which current fairness methods may overlook. We first formally define the intersectional two-sided fairness and conduct empirical experiments to demonstrate the existence of such unfairness and inadequacy of current fairness methods in addressing this problem. To address this problem, we divide the intersectional two-sided fairness into two goals: (i) consistently distinguishing between positives and negatives for different intersectional groups; (ii) fairly treating positives in different intersectional groups. Then, a novel method, ITFR, is proposed to achieve these goals, which consists of *sharpness-aware disadvantage group discovery*, *collaborative loss balance*, and *predicted score normalization*. The first two components aim to achieve the first goal, while *predicted score normalization* is leveraged to achieve the second objective. Extensive experiments on three public datasets show that ITFR effectively alleviates the intersectional two-sided unfairness and consistently outperforms the previous state-of-the-art methods. Further experiments show that our method is also compatible with fairness-aware re-ranking methods. Additionally, to the best of our knowledge, our method is also the first to improve performance-based fairness for both user and item sides.

For future work, we are interested in exploring such intersectional two-sided unfairness at the individual level and exploring better ways to improve the compatibility of fairness methods in the ranking and re-ranking phases.

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

# A APPENDIX

## A.1 Learning Algorithm of Sharpness-aware Collaborative Loss Balance

Algorithm 1 shows the algorithm of sharpness-aware collaborative loss balance.

---

**Algorithm 1** Sharpness-aware Collaborative Loss Balance

---

**Input:** training data $\mathcal{D}$, number of intersectional groups $P \times Q$, learning rate $lr$, hyperparameters $\eta, \gamma, \rho$
**Output:** recommendation model $f(\theta)$

1: initialize recommendation model $f(\theta)$ and group weights $w_{p,q} = \frac{1}{P \times Q}$ for $p = 1, ..., P$ and for $q = 1, ..., Q$.
2: **for** $t = 1$ to $T_{epoch}$ **do**
3:     **for** batch data $\mathcal{D}_{batch}$ in $\mathcal{D}$ **do**
4:        **for** $p = 1$ to $P$ **do**
5:           **for** $q = 1$ to $Q$ **do**
6:              calculate gradients $\nabla_\theta \mathcal{L}_{p,q}(\mathcal{D}; \theta)$ of $g_{p,q}$
7:              calculate sharpness-aware loss $\hat{\mathcal{L}}_{p,q}$ based on Eq.(3)
8:              calculate sharpness-aware gradients $\nabla_\theta \hat{\mathcal{L}}_{p,q}$
9:           **end for**
10:        **end for**
11:        calculate group weight $\beta_{p,q}$ for any $p, q$ based on Eq.(??)
12:        calculate $C(g_{p,q} \to g_{a,b})$ for any $p, q, a, b$ (Eq.(5))
13:        calculate $C(g_{p,q})$ for any $p, q$ based on Eq.(6)
14:        update group weights $w_{p,q}$ for any $p, q$ based on Eq.(7)
15:        update $\theta \leftarrow \theta - lr * \left( \sum_{p,q} w_{p,q} \cdot \nabla_\theta \hat{\mathcal{L}}_{p,q} \right)$
16:     **end for**
17: **end for**
18: **return** recommendation model $f(\theta)$

---

## A.2 Implement Details

We use the classic MF [27] as user and item encoders for all the methods. The embedding size is set to 64 for all the methods. Adam [19] is used as the optimizer. The learning rate is set to 1e-3 with the L2 regularization tuned in [0, 1e-7, 1e-6, 1e-5, 1e-4]. The batch size is set to 1024. For fair comparisons, the uniform negative sampling is applied to all the models during training, except for FairNeg mixed with a fair negative distribution. The negative sampling number for

training is set to 1. The additional hyperparameters for the baselines are fine-tuned following their original paper. The best models are selected based on the performance of the validation set within 200 epochs. We repeat each experiment 5 times and report the average results, and perform statistical tests (i.e., t-test).

