# OpenReview forum: "Intersectional Two-sided Fairness in Recommendation"
_ACM.org/TheWebConf/2024/Conference — TheWebConf24 Oral_

### Official Review · Reviewer_zRk1 · 2023-11-08

**Novelty:** 6
**Technical Quality:** 5

**Review:**

In this paper, the author proposed a new issue called Intersectional Unfairness for the existing fair recommendation field. Compared to other baselines, it can minimize the utility differences among different intersectional groups while maintaining state-of-the-art overall performance.

Pros:
- propose a novel question in the two-sided fair recommendation.
- the empirical study is easy to understand and does a good job of explaining why this work is necessary.

Cons:
- The definition of intersectional group and intersectional unfairness is not explicitly provided in the manuscript. As a key concept of this paper, readers should have a clear picture of them at Sect. 1, or they may get confused.
- The reason for tackling such unfairness is not adequately described. Actually, I think the reasons between line 84 - 93 can be applicable to any two-sided fairness paper. I think the authors should think more about how to present the motivation.

**Questions:**

1. As the authors mentioned diversity between line 84 - 93, did the authors design specific components in the algorithm that can improve the recommendation diversity? If so, is there a directly can we observe the improvement in diversity in the experiments?
2. Can authors provide some insights into why the proposed algorithm can outperform BPR in terms of accuracy?
3. I am interested in the relationship between accuracy and fairness. Considering authors only compare one conventional model BPR and it is quite old (proposed in 2012), can authors provide more results on the newer conventional recommendation models (e.g., LightGCN)? I am curious how the fairness metrics will change with accuracy metrics.

**Reviewer Confidence:**

4: The reviewer is certain that the evaluation is correct and very familiar with the relevant literature

**Scope:**

4: The work is relevant to the Web and to the track, and is of broad interest to the community

---

### Official Review · Reviewer_f6zc · 2023-11-15

**Novelty:** 5
**Technical Quality:** 5

**Review:**

Generally, I think the authors proposed an interesting problem of intersectional fairness for two-sided fairness in recommender system.
* The paper is generally well-written, and motivation is discussed in a decent manner.
* The proposed solution achieves better performance compared to existing approaches. The experiments are quite exhaustive, covering a wide range of scenarios.

However, I have some major concerns regarding this paper, and I am happy to increase the scores if the author can provide clarification or address these concerns.
* One initial concern pertains to the treatment of intersectional fairness in the proposed application; an exploration into the feasibility of transforming the two-sided fairness problem into a single-sided framework consideration. For example, why not dividing the exaxmple in the introduction to Male-Horror, Male-Romance, Female-Horror, Male-Romance and apply single-sided fairness-aware method to solve it? Or at least, such experiment should be included to show why consider it as the intersetional fairness.
* Why choosing ICV and UCV as metrics. Firstly they are not common fairness metric in the literature. Additionally, a nuanced aspect arises in the normalization of sample distances, where the standard deviation of the matrix may appear naturally smaller compared to alternative baselines. Can you help me to clarify it?
* Why choosing BPR loss as the proxy? Firstly, is your method only suitable for the BPR-related model? Secondly, why not choose the metric evaluation as proxy, for example, Demographic Parity or the minimum uility of groups.
* Finally, can you explain why the method can improve the accuracy? And the fairness-aware baseline have comparable performance of BPR? Can you help me to clarify it?

**Questions:**

See review concerns above

**Reviewer Confidence:**

4: The reviewer is certain that the evaluation is correct and very familiar with the relevant literature

**Scope:**

3: The work is somewhat relevant to the Web and to the track, and is of narrow interest to a sub-community

---

### Official Review · Reviewer_YV3v · 2023-11-24

**Novelty:** 4
**Technical Quality:** 5

**Review:**

Pros:

1. The paper works on an important and interesting topic: the fairness of recommendation systems in user and item groups.

2. The paper is well organized and delivered based on the logic of finding the problem, the insufficiency of existing methods, proposing the approach, and experiment validation.

3. The authors provide a comparison of their approach with current recommendation fairness and machine learning intersectional fairness.

4. When introducing the approach, the authors use detailed equations and vivid figures to enhance the paper's readability and technical sound.

Cons:

1. Instead of only displaying the average results, it would be better if the authors could provide the case study of some user/item group, as Table 1 does.

2. Other detailed questions about the experiments. Please see the Questions Section.

**Questions:**

1. In the example of Section 2.3, the user groups are divided by multiple attributes, like race and gender. However, the authors only divide user groups by one attribute (either age or gender). If the user groups are divided by multiple attributes, will the performance gain be consistent?

2. How do the authors deal with the users/items with missing values of the criteria attribute like age or gender? If the information is not provided, does the user/item have to be unfair?

3. Does the approach have the ability to transfer? For example, if the model is trained based on the user groups divided by age, will the user groups divided by gender have worse fairness performance?

**Reviewer Confidence:**

3: The reviewer is confident but not certain that the evaluation is correct

**Scope:**

4: The work is relevant to the Web and to the track, and is of broad interest to the community

---

### Official Review · Reviewer_d7LS · 2023-11-24

**Novelty:** 5
**Technical Quality:** 5

**Review:**

The paper addresses the issue of intersectional two-sided unfairness in recommender systems. To tackle this issue, the authors propose a novel method called Intersectional Two-sided Fairness Recommendation (ITFR). This method uses a sharpness-aware loss to identify disadvantaged groups, collaborative loss balance to develop consistent distinguishing abilities across different intersectional groups, and predicted score normalization to fairly treat positive recommendations in different groups.

Strength:
- The paper addresses an underexplored yet critical aspect of fairness in recommender systems, emphasizing the importance of intersectional two-sided fairness.
- The proposed ITFR method is comprehensive, addressing the issue from multiple angles.
- The authors provide empirical analysis demonstrating the existence of intersectional unfairness

Weakness:
- The definition of intersectional two-sided fairness could be explained more clearly. A mathematical definition may help.
- It is unclear about the intersectional distribution in the datasets used for experiments. The authors should provide more statistics about this information.

**Questions:**

- In line 245, $V_j$ is undefined.
- In Algorithm 1 line 11, the equation is unclear.

**Reviewer Confidence:**

3: The reviewer is confident but not certain that the evaluation is correct

**Scope:**

3: The work is somewhat relevant to the Web and to the track, and is of narrow interest to a sub-community

---

### Official Review · Reviewer_5yn3 · 2023-11-26

**Novelty:** 5
**Technical Quality:** 6

**Review:**

This paper proposes various loss functions to deal with unfairness in both user and item sides.

The paper is well-motivated.
Empirical studies that probe the unfairness are provided in detail.
The experiments are pretty comprehensive.

**Questions:**

N/A

**Reviewer Confidence:**

2: The reviewer is willing to defend the evaluation, but it is likely that the reviewer did not understand parts of the paper

**Scope:**

4: The work is relevant to the Web and to the track, and is of broad interest to the community

---

### Decision · Program_Chairs · 2024-01-22

**Decision:**

Accept (Oral)

**Comment:**

The paper addresses an important problem of intersectional two-sided fairness. The paper is well motivated, and the proposed solution is generally interesting and novel. The experiments are comprehensive to support main claims.

 There are some technical details, particularly the precise definition of intersectional group / intersectional unfairness, should be clarified in the camera ready.